# Diagnosis and Management of Autoimmune Hemolytic Anemia in Patients with Liver and Bowel Disorders

**DOI:** 10.3390/jcm10030423

**Published:** 2021-01-22

**Authors:** Cristiana Bianco, Elena Coluccio, Daniele Prati, Luca Valenti

**Affiliations:** 1Department of Transfusion Medicine and Hematology, Fondazione IRCCS Ca’ Granda Ospedale Maggiore Policlinico, 20122 Milan, Italy; cristiana.bianco@policlinico.mi.it (C.B.); elena.coluccio@policlinico.mi.it (E.C.); daniele.prati@policlinico.mi.it (D.P.); 2Department of Pathophysiology and Transplantation, Università degli Studi di Milano, 20122 Milan, Italy

**Keywords:** autoimmune hemolytic anemia, chronic liver disease, inflammatory bowel disease, autoimmune disease, autoimmune hepatitis, primary biliary cholangitis, treatment, diagnosis

## Abstract

Anemia is a common feature of liver and bowel diseases. Although the main causes of anemia in these conditions are represented by gastrointestinal bleeding and iron deficiency, autoimmune hemolytic anemia should be considered in the differential diagnosis. Due to the epidemiological association, autoimmune hemolytic anemia should particularly be suspected in patients affected by inflammatory and autoimmune diseases, such as autoimmune or acute viral hepatitis, primary biliary cholangitis, and inflammatory bowel disease. In the presence of biochemical indices of hemolysis, the direct antiglobulin test can detect the presence of warm or cold reacting antibodies, allowing for a prompt treatment. Drug-induced, immune-mediated hemolytic anemia should be ruled out. On the other hand, the choice of treatment should consider possible adverse events related to the underlying conditions. Given the adverse impact of anemia on clinical outcomes, maintaining a high clinical suspicion to reach a prompt diagnosis is the key to establishing an adequate treatment.

## 1. Introduction

Anemia is a common feature of hepatic and bowel disorders. Blood loss from the gastrointestinal tract due to portal hypertension and mucosal disease together with chronic inflammation represent the main causes, but autoimmune hemolytic anemias (AIHAs) should also be considered in the differential diagnosis. Though hemolytic anemias may be associated with advanced liver disease or coexistent genetic conditions, AIHA is commonly observed in association with some inflammatory disorders affecting the liver and the gut. In this study, we will review the epidemiology of AIHAs, and the specific challenges related to their diagnosis and treatment in patients with cirrhosis and bowel disorders.

## 2. Anemia in Liver Disease

Anemia is commonly found in patients with chronic liver disease [1,2,3]. In the setting of advanced liver disease, lower hemoglobin levels predict adverse outcomes, including hepatic decompensation [3,4], development of acute on chronic liver failure (ACLF) [3,5], and mortality in patients with hepatocellular carcinoma [3,6]. The main causes of anemia during advanced liver disease are reported in Table 1.

In cirrhotic patients, gastrointestinal bleeding is a common complication of portal hypertension [3,7]. It can have an acute presentation with hematemesis and melena that requires urgent treatment, but it should also be suspected in case of microcytic anemia or a positive fecal occult blood test. Although varices can occur everywhere in the gastrointestinal tract, gastroesophageal varices are the most clinically significant, since their rupture is responsible for about 70% of bleedings [8]. Moreover, cirrhotic patients frequently develop iron deficiency anemia due to chronic blood loss from gastroesophageal varices and hypertensive gastropathy [9,10].

Hypersplenism, in addition to portal hypertension and splenomegaly, can cause hemolytic anemia in patients with chronic liver disease. Typically, hypersplenism is associated with pancytopenia [11,12], and platelets are the main cell type targeted for sequestration and destruction in the spleen [2].

Spur cells are large red blood cells with spikelike projections [13,14]. Spur cell anemia is an uncommon though severe, life-threatening form of anemia in patients with severe liver disease, and manifests with rapidly progressive hemolytic anemia and the presence of acanthocytes in the blood smear [13,15,16]. The change in morphology is due to an imbalance of the cholesterol/phospholipids ratio in the red cell membrane [17,18], and leads to an impaired deformability of erythrocytes and a reduction of cell survival. The presence of spur cell anemia is associated with a poor prognosis [15], and only liver transplantation is considered a curative treatment for the condition [19,20,21].

## 3. Anemia in Gastrointestinal Disease

In pathological conditions affecting the gastrointestinal tract, anemia is most frequently related to blood loss, chronic inflammation, and malabsorption.

Iron deficiency anemia often originates from chronic gastrointestinal blood loss [22]. Therefore, patients with microcytic anemia, low levels of ferritin, and transferrin saturation should be investigated for occult blood loss [23,24,25,26,27]. The use of aspirin or non-steroidal anti-inflammatory drugs (NSAIDs) should be investigated in the clinical history [28,29].

Anemia is also a typical feature of autoimmune gastritis, and it can be the first sign that leads to diagnosis. During autoimmune gastritis, parietal cells are damaged and the secretion of intrinsic factor and acid are suppressed; this results in the impairment of the absorption of vitamin B12 and iron. Pernicious anemia due to vitamin B12 deficiency can also be preceded by milder hematological alterations, including isolated mean corpuscular volume alterations and anisocytosis [30,31]. In the case of a concomitant deficiency of B12 vitamin and iron, anemia is characterized by normal mean cell volume and anisocytosis [32].

## 4. Hepatic and Gastrointestinal Disorders Predisposing to Autoimmune Hemolytic Anemia

Gastrointestinal disorders that are epidemiologically associated with AIHA are reported in Table 2, and the main ones are described below.

### 4.1. Autoimmune Hepatitis

Autoimmune hepatitis (AIH) is characterized by the presence of non-organ-specific circulating autoantibodies and hypergammaglobulinemia. According to seropositivity, AIH is classified as type 1 (anti-nuclear, ANA, and/or anti-smooth-muscle antibodies, (anti-SMA)), which can begin at any age, and as type 2 (anti-liver/kidney microsomal, (anti-LKM), and/or antibodies against liver cytosol (anti-LC1)), which usually affects children and young adults [33]. Association with other autoimmune conditions is not uncommon [34].

The combination of AIH and AIHA/Evans syndrome has consistently been reported in pediatric and adult patients [35,36]. AIH-associated AIHA can be triggered by viral infections such as Parvovirus B19 [37] and hepatitis A virus (HAV) [38]. Jarasvaraparn et al. described the case of a child who developed Evans syndrome before the onset of AIH, which flared up at the time of liver disease presentation [39]. Tokgoz et al. reported the concomitant presentation of Evans syndrome with AIH and nephrotic syndrome in a 12-year-old child affected by CD3ɣ deficiency [40].

In these patients, AIHA was successfully treated with glucocorticoids [35,36,37], while rituximab was effectively and safely administered to patients with Evans syndrome [39,41]. Although the beneficial impact of rituximab in AIH is very controversial [42], Carey et al. reported the case of a 44-year-old woman who developed Evans syndrome concomitantly with an AIH flare despite prednisone and mycophenolate therapy: rituximab administration was followed by remission [43]. 

### 4.2. Primary Biliary Cholangitis

Primary biliary cholangitis (PBC) is a chronic and progressive autoimmune cholestatic liver disease of the small intrahepatic bile ducts, associated with the serologic reactivity to antimitochondrial antibodies (AMA) or specific anti-nuclear antibody (ANA) [44]. PBC is frequently associated with other autoimmune disorders, including AIHA [45,46,47,48], because of a shared genetic susceptibility across the spectrum of organ-specific autoimmune conditions [49]. The prevalence of AIHA in PBC patients is variable. In a large international cohort of 1554 patients with PBC, AIHA was diagnosed in 0.2% [50]. In a retrospective study of 71 patients with AIH/PBC overlap syndrome, 43.6% had an extrahepatic autoimmune manifestation and 1.4% developed AIHA [51]. Among 565 hospitalized patients with primary Sjögren’s syndrome, concomitant PBC was a risk factor for AIHA [52]. 

Immune-mediated or autoimmune anemia, including AIHA as well as pernicious anemia and celiac disease, should therefore be ruled out in patients with PBC and fatigue [49]. Clinicians should pay attention to bilirubin, which is both a key prognostic predictor [53] and a marker of hemolysis. A sudden rise in bilirubin levels in these patients imposes screening for associated hemolysis. Indeed, secondary AIHA is usually responsive to treatment, while on the other hand, a failure to acknowledge hemolysis could lead to the misclassification of PBC severity [54].

Treatment of PBC-associated AIHA encompasses corticosteroids to control the acute phase and eventually an immunosuppressant for maintenance therapy [55]. Ursodeoxycholic acid (UDCA) should be continued, since it represents the first-line treatment for PBC [44]. There is only one report of mild anemia which resolved spontaneously following UDCA [56]. AIHA can also develop during the follow-up in patients on UDCA [57,58]. Karibori et al. reported a case of liver transplant where splenectomy proved curative for a PBC-associated AIHA [59], whereas Retana et al. described three cases of AIHA occurring several years after a liver transplant for PBC despite immunosuppressant therapy [60]. In two cases, hemolysis was successfully treated with steroid and with rituximab as the second-line treatment, while splenectomy was necessary in one [60].

### 4.3. Primary Sclerosing Cholangitis

Primary sclerosing cholangitis (PSC) is a chronic and progressive cholestatic liver disease, characterized by the inflammation and fibrosis of the biliary ducts, determining multifocal biliary strictures [61]. PSC clusters with AIH, inflammatory bowel disease (IBD), and sometimes PBC, but it is rarely associated with other immune-mediated diseases [62].

Only a few cases of PSC associated with AIHA have been described [63], and almost all of these were responsive to corticosteroids [64,65,66].

It is estimated that about 70% of patients with PSC develop IBD (ulcerative colitis being the most frequent) and about 5% of patients with IBD have PSC [67]. Only a few cases of the triad of PSC, IBD, and AIHA have been reported in the literature [68,69,70], and one of these was a child [71].

### 4.4. Inflammatory Bowel Diseases

Ulcerative colitis and Crohn’s disease represent the two main forms of IBD, which includes a group of diseases characterized by the presence of chronic inflammation and damage of the gut [72,73]. Anemia is probably the most common systemic manifestation of IBD. In the majority of cases, the pathogenesis is related to intestinal bleeding, chronic inflammation, and malnutrition: impaired iron, vitamin B12 and folic acid uptake, and inadequate nutrition [74,75]. However, autoimmune etiology should be considered as well, although the prevalence of AIHA in IBD is low [76,77,78]. 

AIHA is more frequently described in patients with active ulcerative colitis, and it is reportedly associated with the activity and the extension of the disease [76,79]. Association between AIHA and Crohn’s disease seems to be rarer, although possible [80,81].

Corticosteroids are the first-line therapy of AHIA associated with IBD, but other immunomodulators/immunosuppressants have also produced beneficial effects in both conditions [82,83]. AIHA reportedly remits with control of IBD [84,85]. In several cases, AIHA remission was induced by curative surgical resection for IBD [76,86,87,88,89]. However, drug-induced hemolysis in patients treated for IBD should be considered, since isolated cases of drug-induced anemia caused by sulfasalazine [90] and infliximab [91,92,93] have been reported.

### 4.5. Viral Hepatitis

#### 4.5.1. Hepatitis C Virus (HCV) Infection Treatment

HCV is a blood-borne virus responsible for a systemic disease. Hepatic manifestations are related to acute and chronic hepatitis, cirrhosis, and hepatocellular carcinoma; extrahepatic manifestations include, among others, cryoglobulinemia, type 2 diabetes, marginal zone lymphoma (which can also be associated with AIHA independently of the presence of HCV infection), and kidney disease [94]. Since HCV infection determines chronic activation and dysregulation of the immune system, HCV-related immune disorders are not uncommon [95]. 

Cryoglobulins consist of two or more (mixed) immunoglobulin isotypes, with (type II) or without (type III) a monoclonal component [96]. Typically, cryoglobulins are insoluble at temperatures < 37 °C and dissolve after rewarming. Mixed cryoglobulins can be detected in 25–30% of patients with HCV infection and determine a cryoglobulinemic vasculitis in about 10–15% of them [96]. AIHA is not frequent during cryoglobulinemic vasculitis, but it has been described [97,98]. Other researchers described the association of warm AIHA with HCV infection in treatment-naïve patients [98,99,100]. 

Although corticosteroids may enhance viral replication, they have been used successfully to treat AIHA associated with HCV [101,102]. In some cases, treatment with other immunosuppressive drugs was needed [98,103]. Etienne et al. reported the case of an elderly man with a history of thrombocytopenic purpura treated ineffectually with corticosteroids, immunoglobulins, splenectomy, and cyclosporin about 20 years before, who developed cold AIHA and was effectively treated with rituximab [104].

Before the direct-acting antiviral agent (DAA) treatment era, AIHA was observed as an adverse event of interferon-based treatment [105,106,107,108], which could complicate drug-induced hemolytic anemia induced by ribavirin [109,110,111]. 

#### 4.5.2. Hepatitis E Virus (HEV) Infection

HEV infects humans through the fecal-oral route, or through the consumption of contaminated food (undercooked animal products, shellfish, etc.) [112,113]. Transmission through blood transfusion, blood-derived products, and solid organ transplant is less common [114,115,116]. In addition to acute and chronic hepatitis, HEV infection is associated with several extrahepatic manifestations, including anemia [117]. Most cases of anemia during HEV infection are due to hemolysis secondary to glucose-6-phospate dehydrogenase deficit [118,119,120,121,122,123,124]. However, a few cases of AIHA have been described both in adults and children [125,126,127]. Aplastic anemia should also be considered in the differential diagnosis [128,129].

Steroid use during HEV infection may not be recommended, due to the possibility of disease chronicization in immunosuppressed individuals. In most reported cases, AIHA was successfully managed by supportive therapy.

#### 4.5.3. HAV Infection

The transmission of HAV is through contaminated food or water, although horizontal transmission is also possible [130]. The presentation of HAV infection ranges from the complete absence of symptoms (mostly in children) to acute/fulminant hepatitis; gastrointestinal symptoms, fever, and malaise are common [131]. 

Anemia of several etiologies is frequently observed during the disease [132]. AIHA has been described during HAV infection [38,133,134,135] and may be associated with red cell aplasia [136,137]. 

In the majority of cases, treatment with steroids is the first choice when the disease does not remit with the cessation of viral replication [133,135,137]. Lyons et al. reported the case of a young woman with AIHA during acute HAV ineffectively managed with supportive therapy, who had to be treated with steroids because of the persistence of hemolysis after five weeks [138]. Chehal et al. described a case of AIHA and red cell aplasia requiring immunoglobulin infusion and cyclosporin administration, in addition to blood transfusion and steroids [136].

### 4.6. Celiac Disease

Celiac disease is a chronic immune-mediated condition affecting the small intestine, induced by dietary gluten in genetically predisposed individuals [139]. Abdominal pain, diarrhea, and weight loss are the most common symptoms, but atypical extraintestinal manifestations are not rare [140]. 

Anemia is a common feature of celiac disease. It is most frequently due to iron deficiency secondary to iron malabsorption [141,142], but folate/vitamin B12 deficiency, blood loss, and inflammation also have a role in its pathogenesis [143,144].

Only a few cases of AIHA/Evans syndrome have been reported during celiac disease [145,146,147,148,149], and in one case steroid therapy could not be prescribed because of concomitant advanced liver disease [150].

## 5. Diagnostic and Therapeutic Challenges Related to Autoimmune Hemolytic Anemia

Although AIHA should be considered in patients with liver disease in the context of the underlying disease, clinicians should pay attention to other mechanisms of hemolytic anemia. Several conditions in the spectrum of liver and bowel disorders represent a risk factor for AIHA. Common signs and symptoms of AIHA, such as fatigue, pallor, jaundice, shortness of breath, confusion, peripheral edema, and splenomegaly, could pass unnoticed in patients with decompensated cirrhosis. For instance, in patients with decompensated cirrhosis associated with severe alcohol abuse, or specific disorders such as Wilson disease, hemolytic anemias could be induced via oxidative stress. Increased levels of conjugated bilirubin due to intrahepatic cholestasis play a role in determining this condition [151].

Laboratory evidence of hemolysis associated with anemia should be evaluated. The baseline assessment should include a complete blood and reticulocyte count, lactate dehydrogenase, indirect bilirubin, haptoglobin levels, and the evaluation of a peripheral blood smear. The autoimmune nature of anemia is based on the demonstration of an immune response directed against autologous red blood cell (RBC) antigens. The direct antiglobulin test (DAT) is used to determine the presence of immunoglobulins and/or complement bound to RBC. Definition of the type and the activity temperature of the autoantibodies bound to RBCs influences the severity of the clinical manifestation and the therapeutic approach (Table 3). In the majority of cases, AIHA is due to the presence of IgG antibodies that are active at 37 °C (wAIHA) and determine extravascular RBC lysis; in the cold forms (cAIHAs), which are typically more severe although rarer, hemolysis is caused by IgM antibodies that react with RBCs at 4 °C and activate complement, often causing intravascular hemolysis [152]. 

The DAT tube test is the gold standard, but in specific cases more sensitive techniques such as microcolumn, solid-phase, flow cytometry, and mitogen-stimulated DAT should be performed [153]. Note that AIHA is still possible in the presence of a negative DAT test: this represents a challenging situation because the diagnosis and the management can be delayed. On the other hand, in some liver conditions associated with hypergammaglobulinemia or paraproteins (e.g., hepatitis virus infection), the DAT test may yield false-positive results. 

Furthermore, the diagnostic workup has to consider some specific aspects of liver disease, which include, for instance, the impact of liver dysfunction on circulating haptoglobin levels. It is decreased or undetectable in hemolysis of any kind, but low haptoglobin levels are observed in liver disease because its production is impaired [154]. When cholestasis or liver failure are associated with anemia, the diagnostic relevance of bilirubin is less helpful; in this case, clinical and laboratory history help to distinguish between an acute and a chronic event.

Finally, it is necessary to collect a detailed clinical history to identify possible drug-induced AIHA. Some drugs used for the treatment of liver and bowel diseases (e.g., antibiotics, or beta blockers used for the treatment of gastroesophageal varices) may be responsible for AIHA [155,156]. Drugs can act with two different mechanisms: by triggering the production of drug-dependent antibodies or drug-independent antibodies. In the suspicion of drug-related AIHA the laboratory workup must be performed in a reference laboratory.

In most cases, treatment of secondary AIHA encompasses the treatment of the underlying disease, or the discontinuation of the use of drugs inducing anemia [157], but the specific features of each case have to be considered. Concerning pharmacological treatment, the first-line approach for wAIHA is based on the administration of steroids [153], which may, however, increase the risk of cirrhosis decompensation due to fluid retention, and of electrolyte disorders. Conversely, steroids may be indicated for acute alcoholic hepatitis and ACLF. In this setting, however, steroids may increase mortality from severe infections [158]. Furthermore, the pharmacokinetics of prednisone and prednisolone are affected by liver failure, and the dose should be adjusted accordingly [159,160]. Rituximab is used, alone or in combination with steroids, as a second-line strategy in wAIHA and represents the first choice in cAIHA [161]. The use of rituximab in patients with IBD should be carefully considered, since it has been associated with the progression or onset of Crohn’s disease and ulcerative colitis during long-term treatment [162,163].

## 6. Conclusions

AIHA is an uncommon yet insidious complication of hepatic and gastrointestinal disorders that is more frequently associated with acute viral hepatitis, autoimmune, and autoinflammatory conditions. Due to the concomitant presence of other triggers of anemia and immune dysregulation, the diagnostic process may be challenging, and treatment possibilities are hampered by the concurrent comorbidities and the risk of adverse reactions. Given the adverse impact of anemia on clinical outcomes, maintaining a high clinical suspicion to reach a prompt diagnosis is key to establishing an adequate treatment.

## Figures and Tables

**Table 1 jcm-10-00423-t001:** The main causes of anemia other than autoimmune hemolytic anemias (AIHAs) in patients with advanced liver disease.

Acute or chronic blood loss	Gastroesophageal varices Hypertensive gastropathy Gastric vascular ectasia Peptic ulcer
Hemolysis	Spur cell anemia and Zieve’s syndrome Hypersplenism Wilson disease Congenital red blood cells and hemoglobin disorders with iron overload Paroxysmal nocturnal hemoglobinuria
Malnutrition and/or malabsorption	Vitamin B12 deficiency Folic acid deficiency Alcohol abuse

**Table 2 jcm-10-00423-t002:** The main gastrointestinal system disorders associated with AIHA.

Liver	Infective disorders	Mononucleosis–CMV infection HCV chronic infection HEV, HAV acute hepatitis Giant cell hepatitis
	Autoimmune disorders	Autoimmune hepatitis (frequently with Evans syndrome) Primary biliary cholangitis (PBC) Primary sclerosing cholangitis (PSC)
	Immune-mediated conditions	Interferon-alfa and/or ribavirin treatment in HCV Chronic liver failure Liver transplant
Gastrointestinal Tract		Ulcerative colitis Crohn’s disease Celiac disease Gastric and intestinal lymphoma

CMV: cytomegalovirus, HCV: hepatitis C virus, HEV: hepatitis E virus, HAV: hepatitis A virus.

**Table 3 jcm-10-00423-t003:** The main types of AIHA.

Type	Mechanism	DAT
wAIHA (50–70%)	IgG (or IgA)	IgG +/− C3
cAIHA (15–25%)	IgM	C3
Mixed type (8–10%)	IgG, IgM	IgG + C3

wAIHA: warm autoimmune hemolytic anemia, cAIHA: cold autoimmune hemolytic anemia.

## Data Availability

Data sharing not applicable.

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
