# Peer review of "Diagnosis and Management of Autoimmune Hemolytic Anemia in Patients with Liver and Bowel Disorders"

_jcm, 2021, doi:10.3390/jcm10030423_

Round 1

Reviewer 1 Report

The manuscript is not focused on autoimmune hemolytic anemia but refers to all types of anemia in liver and bowel diseases.

The manuscript you suggested to review has the title "Diagnosis and management of autoimmune hemolytic anemia in patients with liver and bowel disorders". In English literature there is a lack of reviews about autoimmune hemolytic anemia in liver and bowel diseases. However, autoimmune hemolytic anemia is common in autoimmune liver and bowels diseases. The reader who wants to search the literature about this topic is interested in difficult cases with specific treatment. Authors refer to reported cases but the whole manuscript is not focused on this. In my opinion, there is no need to explain all types of anemia in different liver and bowel diseases or to describe the nature of the diseases. Thus, the authors should revise the article, summarise the other types of anemia in tables for example and focus on autoimmune hemolytic anemia. 

Author Response

We thank the Reviewer for the comment and for giving us the possibility to clarify our point of view.

In our opinion, a brief description of the main causes of anemia in liver and bowel diseases is useful for the clinician who is not an hepatologist/gastroenterologist and has to manage the anemia in these complicated patients. We have only provided a starting point to investigate the clinical condition, avoiding lengthy discussions about diagnosis and treatment. Our manuscript is strongly focused on the differential diagnosis and treatment of autoimmune hemolytic anemias in liver and gastrointestinal conditions.

As you rightly point out, few works deal with the issue of AIHA associated with hepatic and gastrointestinal diseases, since AIHA is rarely associated with them. Only in few cases it has been possible to specify the prevalence of AIHA (0.2% in primary biliary cholangitis; 25-30% of HCV patients present cryoglobulins, but the minority of them develops AIHA), since the majority of papers are anectodical case reports without any specific detail regarding diagnostic challenge and universally recognized therapeutic approach in this specific setting. On these bases, we tried to offer a view which was comprehensive of what is available in the literature and worthy of consideration for clinicians in different fields who may be involved in managing these patients. We manuscript has now been improved thanks to your suggestions.

Reviewer 2 Report

In this review, Bianco and coauthors describe and discuss the occurrence of autoimmune hemolytic anemia (AIHA) as related to hepatic and gastrointestinal disorders. They focus in particular on diagnosis and therapy. Although of definite clinical relevance, this approach is quite innovative and not often focused in review articles. Therefore, this paper should be of considerable interest for hematologists, gastroenterologists, hepatologists and general internists. The discussion is adequately based on the literature cited and the selection of references is appropriate. The article is well edited. I have some suggestions for further improvement, all of which can be considered minor comments.

  1. Section 2 (Anemia in liver disease) is a background section not specifically describing the topic of the article. As such, this part of the text appears somewhat too detailed and might be simplified/abbreviated. The same might apply to Section 3, 3rd paragraph.
  2. Table 1: This list of main causes of anemia in liver disease does obviously not include AIHA. Should this appear from the heading/title of the table?
  3. Title of Section 4 should read: “…predisposing to autoimmune hemolytic anemia.”
  4. Subsection 4.4, 1st sentence: “…characterized by…”: I think this applies to all IBD by definition. Consider rewriting the sentence.
  5. Subsection 4.5.1: The authors correctly list “marginal B-cell lymphoma” among diseases associated with HCV. Comments: (a) The accepted WHO term for this entity is marginal zone lymphoma (MZL). Please correct. (b) MZL by itself may be associated with AIHA of the warm as well as cold antibody type. This might be briefly mentioned.
  6. Section 5, 4th paragraph states that haptoglobin levels are usually “decreased or reduced” in hemolysis. I would prefer “decreased or undetectable”.
  7. Section 5, 5th paragraph, sentence “Treatment should…”: I understand what you mean, but this may be misinterpreted as a treatment recommendation. Consider rewrite.
  8. Section 5, last paragraph, “Treatment of secondary AIHA…”: Although this sentence is correct in the setting described in this article, this statement is not universally true. Consider rewrite.
  9. Although generally good, the English language should be thoroughly reviewed and some improvements should be made. Examples:
    1. Subsection 4.1, very last sentence: Replace “where” by “in whom”.
    2. Subsection 4.2, line 7 from top: Delete “of them”.
    3. Section 5, 2nd paragraph, “To this aim is necessary to perform…”: Please rewrite.

Author Response

We thank the Reviewer for the positive evaluation, constructive suggestions, and for the opportunity to improve our manuscript.

  1. Section 2 (Anemia in liver disease) is a background section not specifically describing the topic of the article. As such, this part of the text appears somewhat too detailed and might be simplified/abbreviated. The same might apply to Section 3, 3rd paragraph.

We are aware of the fact that the main topic of our review is autoimmune hemolytic anemia and that sections 2 and 3 may appear less focused on the main manuscript topic. However, given that the paper is addressed to specialists in different fields, we think that an overview of the main causes of anemia associated with liver and bowel disorders could be helpful in clinical practice, especially for clinician who may be involved in diagnosing this condition. For this reason, we limited to briefly describe the main etiologies and give a starting point for the differential diagnosis.

  1. Table 1: This list of main causes of anemia in liver disease does obviously not include AIHA. Should this appear from the heading/title of the table?

In compiling this list we actually focused on causes of anemia other than AIHA, being AIHA the main topic of the paper and discussed below. As suggested by the Reviewer, we added “other than AIHA” to the title of the table (line 41).

  1. Title of Section 4 should read: “…predisposing to autoimmune hemolytic anemia.”

We thank the Reviewer for the specification, we modified the title of the section accordingly (line 79).

  1. Subsection 4.4, 1st sentence: “…characterized by…”: I think this applies to all IBD by definition. Consider rewriting the sentence.

We changed the sentence as suggested to “Ulcerative colitis and Chron’s disease represent the two main forms of IBD, which include a group of diseases characterized by the presence of chronic inflammation and damage of the gut” (lines 148-149).

  1. Subsection 4.5.1: The authors correctly list “marginal B-cell lymphoma” among diseases associated with HCV. Comments: (a) The accepted WHO term for this entity is marginal zone lymphoma (MZL). Please correct. (b) MZL by itself may be associated with AIHA of the warm as well as cold antibody type. This might be briefly mentioned.

We thank the Reviewer for the remark. We modified the manuscript accordingly (line 170) and we added “(that can also be associated with AIHA independently of the presence of HCV infection)” (lines 170-171).

  1. Section 5, 4th paragraph states that haptoglobin levels are usually “decreased or reduced” in hemolysis. I would prefer “decreased or undetectable”.

We substituted the term “reduced” with “undetectable” (line 269).

  1. Section 5, 5th paragraph, sentence “Treatment should…”: I understand what you mean, but this may be misinterpreted as a treatment recommendation. Consider rewrite.

We modified the sentence in: “Some drugs used for the treatment of liver and bowel diseases (as antibiotics, or beta blockers used for the treatment of gastroesophageal varices) may be responsible of AIHA” (lines 275-277).

  1. Section 5, last paragraph, “Treatment of secondary AIHA…”: Although this sentence is correct in the setting described in this article, this statement is not universally true. Consider rewrite.

We changed the statement to: “In most of the cases treatment of secondary AIHA encompasses the treatment of the underlying disease or the discontinuation of the use of drugs inducing anemia, yet the specific features of each single case have to be considered " (lines 289-291).

  1. Although generally good, the English language should be thoroughly reviewed and some improvements should be made. Examples:
    1. Subsection 4.1, very last sentence: Replace “where” by “in whom”.
    2. Subsection 4.2, line 7 from top: Delete “of them”.
    3. Section 5, 2nd paragraph, “To this aim is necessary to perform…”: Please rewrite.

We thank the Reviewer for those correction and we edited the manuscript accordingly (lines 103; 112; 249-251).

Round 2

Reviewer 1 Report

-

Reviewer 2 Report

In the revised manuscript and response letter, the authors have satisfactorily addressed all comments from this reviewer. The paper now appears as a clinically relevant, high-quality review article and I have no further comments.